# Numerical Analysis of Supersonic Impinging Jet Flows of Particle-Gas Two Phases

**Guang Zhang [1], Guang Fei Ma [2], Heuy Dong Kim [3] and Zhe Lin [1],***

[1]  State-Province Joint Engineering Lab of Fluid Transmission System Technology, Zhejiang Sci-Tech University, Hangzhou 310018, China; zhang1134@zstu.edu.cn
[2]  Standard & Quality Control Research Institute Ministry of Water, Hangzhou 310024, China; maguangfei@126.com
[3]  Department of Mechanical Engineering, Andong National University, Andong-Si 1375, Korea; kimhd@anu.ac.kr
*  Correspondence: linzhe0122@zstu.edu.cn; Tel.: +86-0571-86843348

**Abstract:** Supersonic impinging jet flows always occur when aircrafts start short takeoff and vertical landing from the ground. Supersonic flows with residues produced by chemical reaction of fuel mixture have the potential of reducing aircraft performance and landing ground. The adverse flow conditions such as impinging force, high noise spectrum, and high shear stress always take place. Due to rare data on particle-gas impinging jet flows to date, three-dimensional numerical simulations were carried out to investigate supersonic impinging jet flows of particle-gas two phases in the present studies. A convergent sonic nozzle and a convergent-divergent supersonic nozzle were used to induce supersonic impinging jet flows. Discrete phase model (DPM), where interaction with continuous phase and two-way turbulence coupling model were considered, was used to simulate particle-gas flows. Effects of different particle diameter and Stokes number were investigated. Particle mass loading of 10% were considered for all simulations. Gas and particle velocity contours, wall shear stress, and impinging force on the ground surface were obtained to describe different phenomena inside impinging and wall jet flows of single gas phase and gas-particle two phases.

**Keywords:** supersonic impinging jet; particle-gas flows; stokes number; wall shear stress; impinging force

## 1. Introduction

Supersonic impinging jet flows always occur as aircrafts start taking off or landing near the ground in aerospace engineering fields. Many unexpected phenomena can occur and lead to damages in aircrafts and landing ground. Due to flow entrainment associated with the lifting jets, the lift loss induces low surface pressure on the airframe, which results to a force opposing the lift. The loss increases with the decrease of the distance from aircrafts to landing ground. The impingement at high speed and temperature jets to landing ground leads to significant thermal loading and erosion. Another adverse effect is an acoustic landing caused by high sound pressure levels in supersonic impinging jets.

The schematic of supersonic impinging jet flows is shown in Figure 1. Three flow zones are obviously distinguished as primary jet zone, impinging jet zone, and wall jet zone [1]. Primary jet zone and impinging jet zone are separated by plate shock wave. Supersonic flows and shock wave systems are always induced by supersonic or sonic nozzles in primary jet zone. As nozzle pressure ratio between nozzle inlet pressure and outlet pressure is high enough, the nozzle flow will be choked at the nozzle throat and over-expanded or under-expanded at the nozzle exit. As the pressure at the nozzle exit is lower than the ambient pressure, the nozzle flow will be over-expanded. Oppositely, it will be under-expanded. The impinging jet zone is a region close to the jet impingement layer caused by strong pressure gradient. The stagnation bubbles

and recirculating flows always occur in this region. The wall jet zone develops when supersonic jet flows impinge to the ground plate. Flows from primary jet zone are divided into two streams where the outer flows move into wall jet zone and the inner flows are wrapped into impinging jet zone. Shock wave systems also take place in wall jet zone.

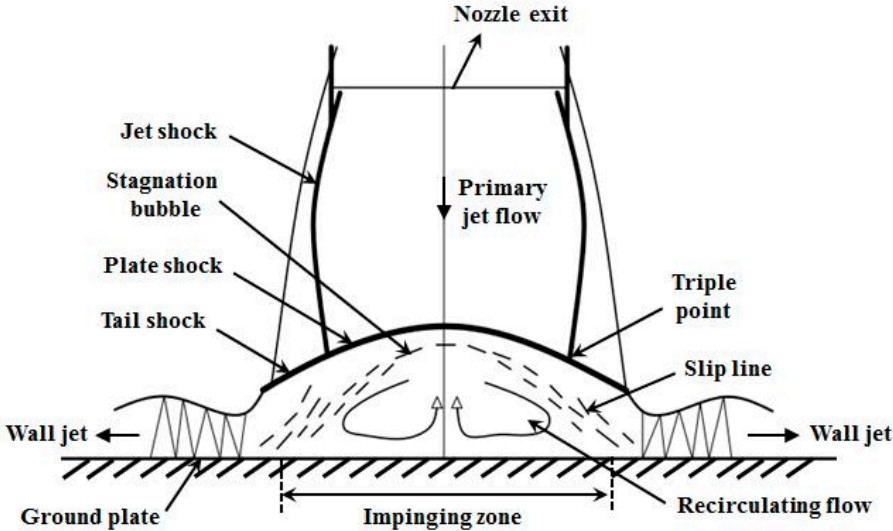

**Figure 1.** Flow field schematic of supersonic impinging jet.

Chemical reaction or combustions occur in combustors when aircrafts start taking off or landing on the ground. Residuals moving from combustors to the exit of rocket nozzle are always liquid drops or solid particles. Solid particles have adverse effects on performance of aircrafts and rockets. Interaction between gas and solid particles results to the loss in momentum and energy of flows. Particle collision and thermal transfer also distribute to this loss and damage nozzle walls of rockets. Therefore, it is significantly important to investigate supersonic impinging jet flows of particle-gas two-phase for improving performance of aircrafts and rockets in aerospace engineering fields.

Experimental studies were carried out to investigate supersonic impinging jet flows induced by supersonic and sonic nozzles respectively. Effects of nozzle pressure ratios and the distance from nozzle exit to ground plate on characteristics of supersonic impinging jet flows were studied. Results showed that high pressure oscillation took place in the plate ground and complex unsteady flow with shock wave and high vorticity regions were also observed [1]. Experimental studies were also performed to investigate the structure of supersonic impinging jet on a large plate by Digital Particle Image Velocimetry (DPIV), Schlieren Visualization, and acoustic measurements. Different distances from nozzle exit to ground plate were tested. Location and strength of plate shock wave were obtained to be different as the distance from nozzle exit to ground plate was varied. Significant oscillations of both compression and expansion regions in the peripheral supersonic flow were also observed [2–5]. Particle Image Velocimetry (PIV) measurements were conducted to study supersonic impinging jet flows. Jet flows were visualized and acoustic spectra levels were obtained to describe impinging jet flows [6]. Different nozzle pressure ratios were varied to study the role of large-scale structures in lift loss, ground erosion, and sonic fatigue for STOVL aircraft in hover at close proximity to the ground by PIV measurements [7]. Wall jet zone was found to be strongly independent of the existence of bubbles occurring in the shock layer [8].

A numerical analysis on oscillatory behaviors of supersonic impinging jet flows was performed and the frequency variation of surface pressure oscillation was investigated. The behavior of flow structure and shear layer between the supersonic and ambient fluids was investigated [9]. Numerical simulations were used to study supersonic impinging jet flows on inclined plates. Stagnation bubbles and jet structures were investigated. Flow structures of free impinging jets were experimentally and

numerically investigated at various impinging angle and nozzle-to-wall distance [10]. Large eddy simulation (LES) and Reynolds-averaged Navier-Stokes (RANS) models were conducted to calculate supersonic impinging jet flows at varying nozzle-to-wall distances and impinging angles. Schlieren Visualization was carried out to compare with numerical results [11]. LES was used to study unsteady and steady supersonic impinging jet flows at different nozzle-to-wall distance as well [12].

Solid particles show different behaviors in the subsonic and supersonic flow regimes. Particles experience sudden acceleration or deceleration when they move through shock waves or expansions. Drag coefficient of spherical particles was calculated in supersonic and subsonic particle-gas flow respectively by using different models [13]. Stokes number and separation of solid particles were investigated in hypersonic flows. Particles calculated at smaller Stokes number were shown to track hypersonic flows more faithfully [14]. A new fluctuating force model was derived to investigate the effect of flow turbulence on solid particles with high Stokes number inducing large particle relaxation time. The change of force distribution and equilibrium diffusion coefficient were proved to be related to time correlation functions of flow velocity fluctuation [15].

Different convergent-divergent nozzles were used to study the velocity and behavior of solid particles. It was found that the particle velocity and behavior were strongly determined by the mixing ratio and the geometry of nozzle inlet and throat [16]. Rudinger et al. carried out theoretical and numerical studies on particle-gas flows through different nozzles. Modified models for calculating supersonic particle-gas flows were derived, which were conducted in numerical simulations [17–19]. Numerical and experimental investigations of multiphase flows were performed and different calculation models of multiphase flows were modified to adjust the real multiphase conditions [20–25]. Numerical simulations were carried out to study multiphase flows in supersonic nozzles. Different drop diameter was investigated to show effects of Stokes number on flow characteristics. Results showed that particles followed supersonic flow more properly at lower Stokes number [26,27].

In the present studies, three-dimensional numerical simulations were used to investigate supersonic impinging jet flows of particle-gas two-phase. Supersonic and sonic nozzles were respectively used to study different supersonic impinging jet flows, particle motion, and shear stress on the landing plate. Particle diameter varied from 1 to 10 μm, which was also described by different Stokes number of particles. Different particle mass loadings were also investigated. Comparison on flow characteristics, impinging force and wall shear stress on the ground plate for single gas phase and particle-gas two phases were discussed in detail.

## 2. Theoretical Analysis

### 2.1. Pressure Coefficient

Pressure coefficient is a key parameter indicating flow characteristics in impinging jet zone and wall jet zone [1]. Recirculating flows and shock wave systems take place in these two zones and pressure changes can be considered to explain flow regimes. Pressure coefficient can be calculated by using Equation (1):

$$C_p = \frac{P_S - P_\infty}{P_0 - P_\infty} \tag{1}$$

where $C_p$ is pressure coefficient on ground plate, $P_s$, $P_0$ and $P_\infty$ are surface pressure, inlet stagnation pressure, and ambient pressure, respectively.

### 2.2. Particle Drag Coefficient

Solid particles behave differently in the subsonic and supersonic flows. Particle drag force experiences sudden acceleration or deceleration due to fluctuations of flow velocity or the presence of shock waves. Drag coefficient of particles strongly depends on particle Reynolds number and flow Mach number. Drag coefficients of spherical particles for supersonic and subsonic flows were derived as shown in following Equations (2) and (3) from Ref. [13].

For subsonic flow,

$$
\begin{aligned}
C_D = &\ 24\left\{\mathrm{Re} + S\left[4.33 + 1.567 \times \exp\left(-0.247\tfrac{\mathrm{Re}}{S}\right)\right]\right\}^{-1} \\
&+ \exp\left(-0.5\tfrac{M}{\sqrt{\mathrm{Re}}}\right)\left[\tfrac{4.5+0.38(0.03\mathrm{Re}+0.48\sqrt{\mathrm{Re}})}{1+0.03\mathrm{Re}+0.48\sqrt{\mathrm{Re}}} + 0.1M^2 + 0.2M^8\right] + \left[1 - \exp\left(-\tfrac{M}{\mathrm{Re}}\right)\right] \times 0.6S
\end{aligned}
\tag{2}
$$

For supersonic flow ($M_g > 1.75$),

$$
C_D = \frac{0.9 + \frac{0.34}{M^2} + 1.86\left(\frac{M}{\mathrm{Re}}\right)^{\frac{1}{2}}\left[2 + \frac{2}{S^2} + \frac{1.058}{S} + \frac{1}{S^4}\right]}{1 + 1.86\left(\frac{M}{\mathrm{Re}}\right)^{\frac{1}{2}}}
\tag{3}
$$

For supersonic flows at Mach number between 1 and 1.75, a linear interpolation is considered and drag coefficient can be expressed by Equation (4) from Ref. [13]:

$$
C_D = \frac{24}{\mathrm{Re}}\frac{1 + \exp\left(-\frac{0.427}{M^{4.63}} - \frac{3}{\mathrm{Re}^{0.88}}\right)}{1 + \frac{M}{\mathrm{Re}}\left[3.82 + 1.28\exp\left(-1.25\frac{\mathrm{Re}}{M}\right)\right]}
\tag{4}
$$

where $C_D$ is drag coefficient and *Re* is particle Reynolds number. $M_g$ is Mach number of gas phase, $M$ is Mach number based on relative velocity between gas phase and particle phase, and $S$ is the molecular speed ratio.

$$
S = M\sqrt{\frac{\gamma}{2}}
\tag{5}
$$

Particle Reynolds number that depends on relative velocity between gas phase and particle phase is obtained by using Equation (6).

$$
\mathrm{Re} = \frac{\rho_P\left|U - U_P\right|D_P}{\mu}
\tag{6}
$$

where $\rho_P$ is particle density and $D_p$ is particle diameter. $\mu$ is dynamic viscosity of fluid and $U$ and $U_p$ are the velocity of gas phase and particle phase, respectively. Based on different expressions of drag coefficients for supersonic and subsonic particle-gas flows, particle drag force $F_D$ can be obtained as shown in Equation (7):

$$
F_D = \frac{3\mu C_D \mathrm{Re}}{4\rho_P D_P{}^2}
\tag{7}
$$

The correlation on drag coefficients was incorporated in Ansys Fluent by user defined function (UDF) for numerical simulations on particle dynamics.

*2.3. Stokes Number*

Stokes number is a non-dimensional parameter which shows the behavior of solid particles suspended in the fluid flow. Stokes number *Stk* is defined as the ratio of characteristic time of particles to characteristic time of gas phase as shown in Equation (8):

$$
Stk = \frac{t_0 U}{L}
\tag{8}
$$

where $t_0$ is relaxation time and $L$ is characteristic length of geometry. Particles at lower Stokes number follow flow streamlines more properly and show better tracing accuracy [27]. Relaxation time is regarded as a measurement of the responsiveness of particles to a change in flow velocity.

$$
t_0 = \frac{\rho_P D_P^2}{(18\mu)}
\tag{9}
$$

In the present studies, the average density of particles is 1550 kg/m$^3$ and nozzle pressure ratio is fixed to be 5. Relaxation time $t_0$ and Stokes number of particles were calculated to be 4.76 μs, 119 μs, 476 μs, and 0.112, 2.8, 11.2 as particle diameter is 1 μm, 5 μm, 10 μm respectively.

### 2.4. Impinging Force on Plate Ground

Impinging force on the ground plate is induced by supersonic particle-gas flows colliding to the ground plate. Particle-gas flows were initialized in high momentum through supersonic or sonic nozzles. One part of flows from the primary jet moves towards the ground plate vertically, which affects impinging force mostly, and the other part of flows moves into wall jet zone. The impinging force can be calculated by the following Equation (10):

$$F_I = P_{AV} \times A \tag{10}$$

where $F_I$ is impinging force on calculated area and $P_{AV}$ is area average pressure on calculated area. $A$ is the calculated area. Due to a large domain used as ground plate, the area where it is greatly affected by the pressure was used for calculating impinging force as shown in Figure 2a,b. Full ground plate was proved to be not suitable for showing the difference in impinging force between single gas flow and particle-gas flows.

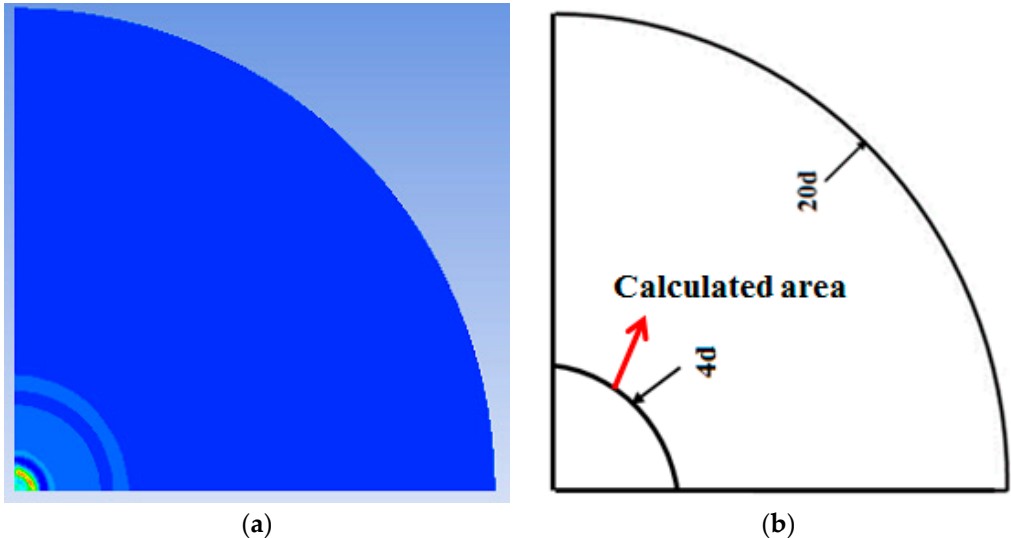

|(a)|(b)|

**Figure 2.** Pressure contours on the ground plate and calculated area for calculating impinging force. (**a**) Pressure contours on the ground plate; (**b**) Calculated area.

## 3. Numerical Methods

### 3.1. Computational Domain

The schematic of computational model is shown in Figure 3. The model was obtained from Ref. [1] where supersonic and sonic nozzles were used to induce different impinging jet flows. In the present CFD studies, three-dimensional sonic nozzle was used and the detailed size was shown in Figure 4. The throat diameter d of the nozzle is 25.4 mm and the designed Mach number is 1.5. The nozzle inlet has the diameter of 1.42d and the convergent section is restricted to be 19.3°. Three-dimensional symmetric domain which was considered by using a quarter of full domain was simulated. h representing distance from nozzle exit to ground plate was fixed to be 3d.

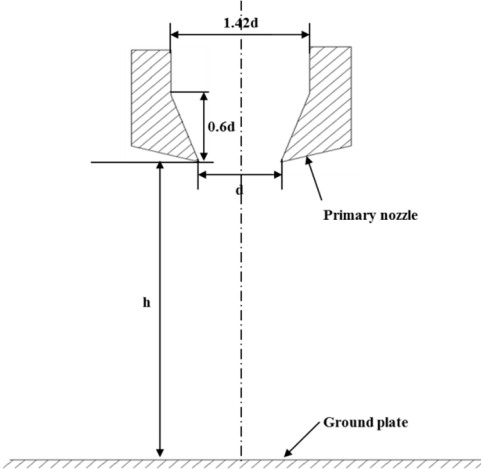

**Figure 3.** Experimental model.

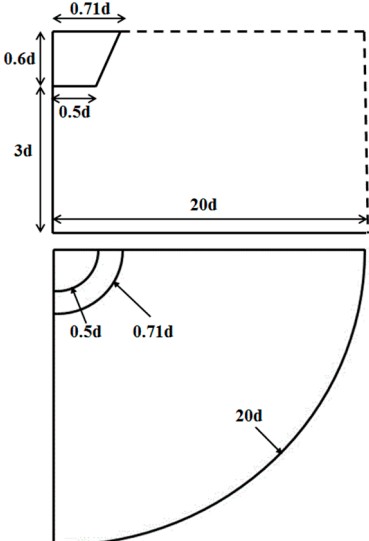

**Figure 4.** Three-dimensional computational domain.

### 3.2. Numerical Schemes

Supersonic jet flows were mathematically analyzed by using unsteady Reynolds-averaged Navier-Stokes (RANS) equations. Turbulence model was solved by k-ω shear stress transport (SST) and Sutherland viscosity model which indicates the gas viscosity changes with flow temperature was used as viscosity model. AUSM scheme was used as the flux model and the second order implicit scheme was used for temporal discretization. Spatial discretization was described by using second order upwind scheme. Structure meshes were created for full computational domain and boundary layer meshes were also used near all walls. Discrete Phase Model (DPM) where Lagrangian–Eulerian track method is considered was used to calculate gas-particle flows. Particle phase was regarded as discrete phase and gas phase was continuous phase. Two-way turbulence coupled interaction was considered to express the interaction between solid particles and gas phase.

### 3.3. Boundary Conditions

Nozzle pressure ratio was fixed to be 5 and the ambient pressure was 0.1 Mpa. Working fluid assumed as ideal gas was initialized in constant total temperature of 293.5 K. Nozzle walls and ground plate were all considered as adiabatic walls with constant temperature of 293.5 K. Detailed boundary conditions of full computational domain are shown in Figure 5. Particles were seeded after the calculation

of single gas flow was convergent. The injection was set at the position of nozzle inlet. Anthracite with the density of 1550 kg/m$^3$ was used as injected particles. Particle mass loading which defines as particle mass flow rate occupying the percent from total mass flow rate of gas-particle flows was fixed to be 10%. Effects of particle diameter varied from 1 to 10 μm on particle-gas flows were investigated.

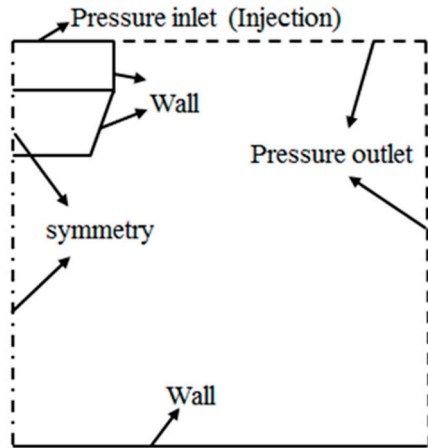

**Figure 5.** Computational domain.

### 3.4. Mesh Independence Study

Grid quality is very important for numerical simulations, and a mesh independence study was carried out to make sure that the present grid quality is good enough for numerical simulations. A supersonic nozzle was simulated at the pressure ratio of 3.5 and h = 3d and pressure coefficients were obtained at different grid numbers for the same computational domain as shown in Figure 6. Shock wave system is greatly different at the grid number of 635,800 compared to other two grid numbers. In order to make sure the precision and save the time of calculation, the grid number of 853,500 was chosen for the present simulations.

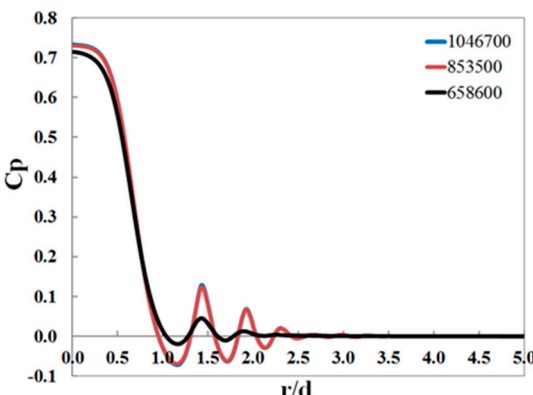

**Figure 6.** Pressure coefficients at different grid numbers.

## 4. Results and Discussion

### 4.1. Validation

In order to validate the accuracy of numerical methods for present simulations, comparisons between experimental and CFD results on pressure coefficients on ground plate were carried out, as shown in Figure 7. In the experimental studies [1], 32 pressure transducers were arranged along a radial line to measure the mean surface pressure on the ground plate. Pressure transducers were installed near the jet centerline more closely due to that the mean pressure variations were more significant. Different operating conditions such as supersonic or sonic nozzles, nozzle pressure ratios,

and distance between the nozzle exit and the ground plate were considered. In the present numerical simulations, turbulence model of k-ω SST was used, which has the advantage on calculating wall bounded problems and jet flows. Supersonic nozzle was used to induce impinging jet flows at nozzle pressure ratio of 5 and h = 3d. Pressure coefficients calculated from the numerical simulation were shown to agree with experimental results well. This indicates that present numerical methods can be used to simulate supersonic impinging jet flows.

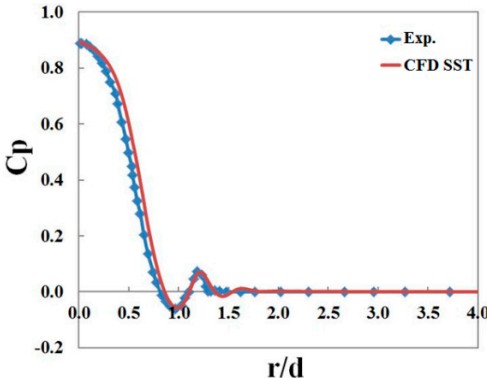

**Figure 7.** Comparison between experimental and CFD pressure coefficients.

## 4.2. Particle-Gas Flows Induced by Different Nozzles

Supersonic and sonic nozzles were respectively investigated at the pressure ratio of 5 and h = 3d. Particle and flow velocity contours are shown in Figure 8a–d. Compared to flow velocity contours, particle distributions and slight velocity difference between gas and particle phase indicated that particles tracked supersonic flows properly. Primary jet zone, impinging jet zone and wall jet zone were obviously observed. Nozzle flows were under-expanded at the nozzle exits for both nozzles. The plate shock wave almost parallel to the ground plate was clearly observed in impinging jet zone. Supersonic flows in the wall jet zone indicated weak shock systems took place in this zone. The maximum flow velocity induced by the sonic nozzle was higher than that induced by the supersonic nozzle due to effects of solid particles. Particles affect supersonic flows more greatly when flow velocity is higher and shock wave system is more complex. The reason why flow velocity is higher in sonic nozzle is that solid particles affect flows more greatly in supersonic nozzle due to larger inertia and resistance. The maximum particle velocity was also observed to be higher by using the sonic nozzle. Particle velocity was obtained to be slightly lower than gas velocity, which mainly results from the inertia and resistance of particles in supersonic flows. The slight difference between gas and particle velocity also showed particles used for present simulations followed subsonic and supersonic flows well by interpolating drag coefficient models from Henderson [13].

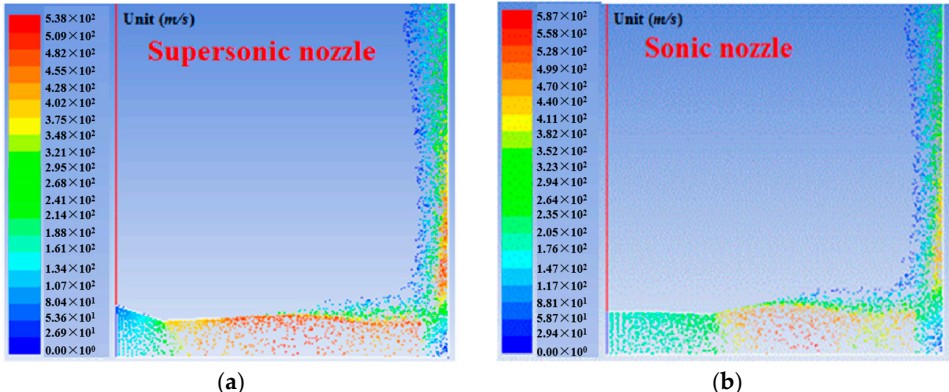

(**a**)　　　　　　　　　　　　　　　　　　　　　　(**b**)

**Figure 8.** *Cont.*

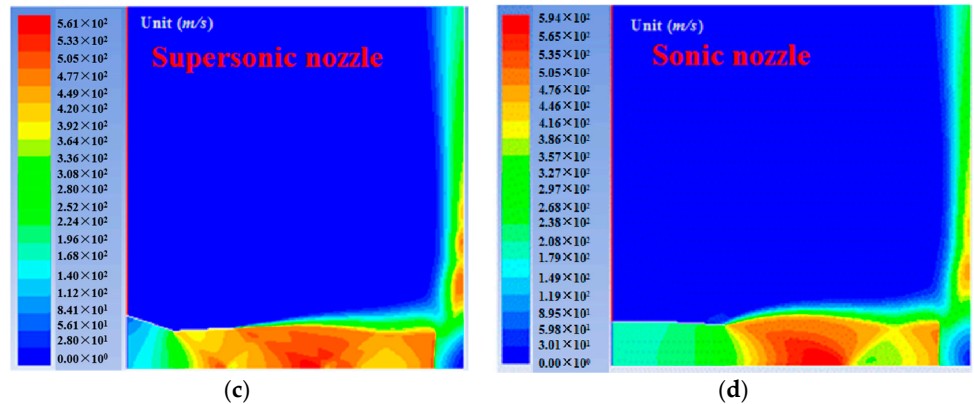

**Figure 8.** Velocity magnitude contours of particle and gas flow. (**a**) Particle velocity; (**b**) Particle velocity; (**c**) Gas velocity; (**d**) Gas velocity.

Pressure coefficients on the ground plate induced by supersonic and sonic nozzles were calculated as shown in Figure 9. Different flow characteristics occurred in imping jet zone and wall jet zone. In the impinging jet zone, recirculating flow region where the pressure dropped was observed by using sonic nozzle. In addition, supersonic flow regions in the wall jet zone induced by sonic nozzle were obtained to be smaller compared to that induced by supersonic nozzle. In the wall zone, the strength of shock waves was shown to be stronger by using supersonic nozzle, which was explained by higher amplitude of pressure changes.

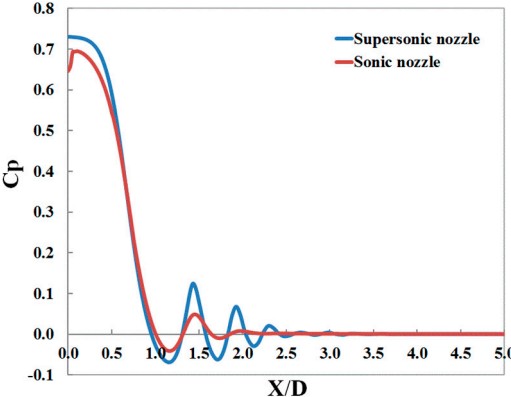

**Figure 9.** Pressure coefficients on the ground plate by using different nozzles.

### 4.3. Effect of Stokes Number

Particle velocity contours at different Stokes number are shown in Figure 10a–d. Particle diameters of 1 μm, 5 μm, and 10 μm were tested, which led to Stokes number of particles to be 0.112, 2.8, and 11.2 respectively. A particle diameter of 0 μm means no particles were injected into the flow. Particles with the diameter of 1 μm tracked the flows well while particles became disordered especially in wall jet zones as Stokes number of particles gradually increased. As previously mentioned, smaller Stokes number represents better tracing accuracy. As Stokes number of particles increased, the recirculating flow region inside imping jet zone gradually became smaller and disappeared at Stokes number of 11.2. Particles inside wall jet zones became more disordered and some particles moved outside of wall jet zones. As the particle diameter increased, particle velocity gradually decreased, which is due to that larger particles need more momentum to be accelerated.

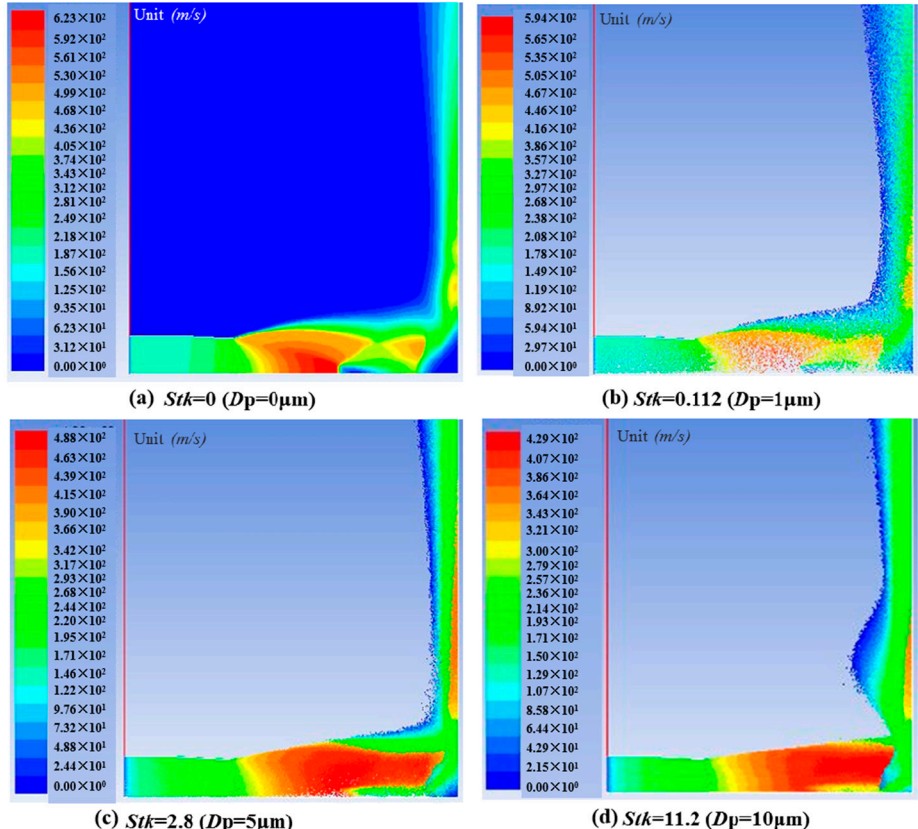

**Figure 10.** Particle velocity contours at different Stokes numbers.

### 4.4. Particle Effects

Three-dimensional sonic nozzle was used to initialize supersonic impinging jet flows. The nozzle pressure ratio was 5 and particle diameter was kept constant of 1 μm. Particle mass loading was fixed to be 10%. Mach number contours for single gas flow and particle-gas flows are shown in Figure 11a,b. In single gas flow, nozzle flows were under-expanded at given pressure conditions. Expansion and reflected shock waves were observed in primary jet zone. Strong plate shock wave and stagnation bubble were clearly observed. However, for gas-phase flows, even though nozzle flows were also under-expanded, flows inside impinging jet zone showed significant different behaviors. The plate shock wave disappeared and a large recirculating flow region occurred instead of the previous bubble. Flow velocity contours were obtained at two planes of XOY and XOZ as shown in Figure 12a,b and Figure 13a,b. Flows were shown similar characteristics in both planes. Based on flow streamlines, most flows moved into wall jet zone.

The other flows moved towards impinging jet zone and recirculated back to primary jet zone for particle-gas flows. High shear stress layer happened at outside of the recirculating flow region. Particle-gas flows followed the shear stress layer and moved into wall jet zone. Flow Mach number was observed to be much lower for particle-gas flows. This mainly resulted from the interaction between gas phase and particle phase. In CFD studies, interaction with continuous phase and two-way turbulence coupling model were considered. Particles attenuates gas phase due to inertia and resistance, and oppositely gas phase also affects discrete phase. Particle collision and heat transfer between two phases also led to more decay in momentum and energy of flows.

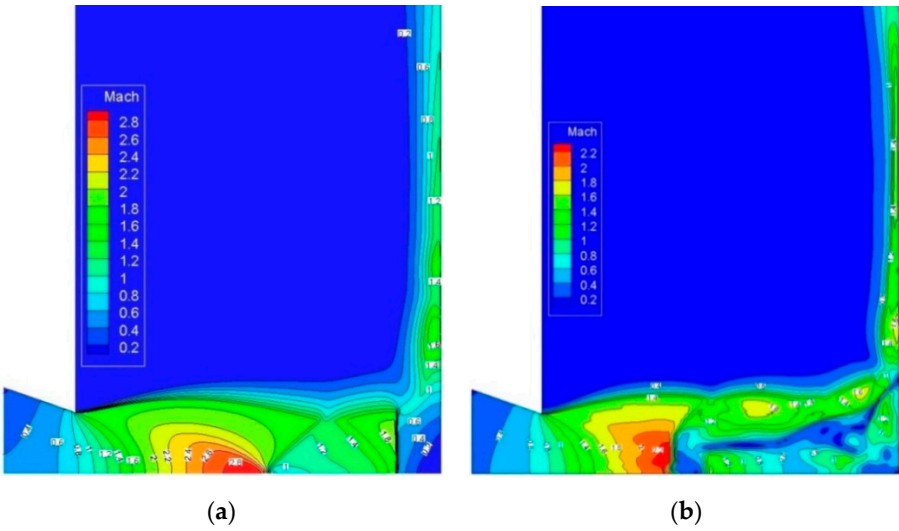

**Figure 11.** Mach number contours for single gas flow and gas-particle flows. (**a**) Single gas flow; (**b**) Gas-particle flows.

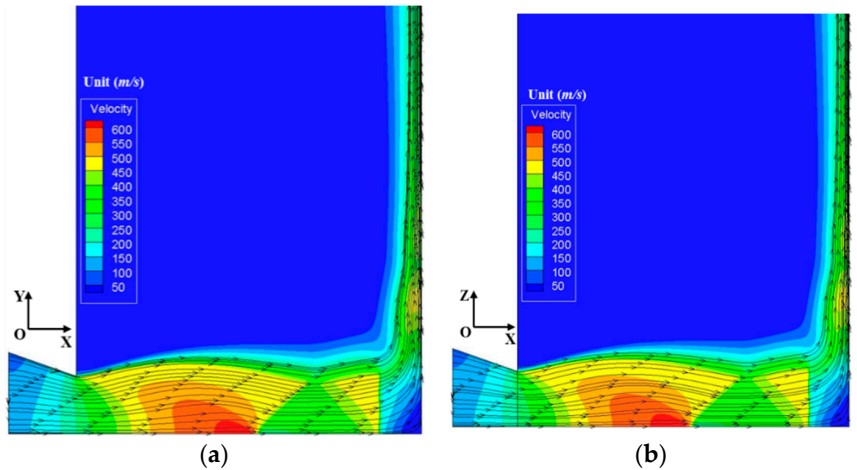

**Figure 12.** Velocity contours and streamlines at different cross-sections for single gas flow. (**a**) XOY plane; (**b**) XOZ plane.

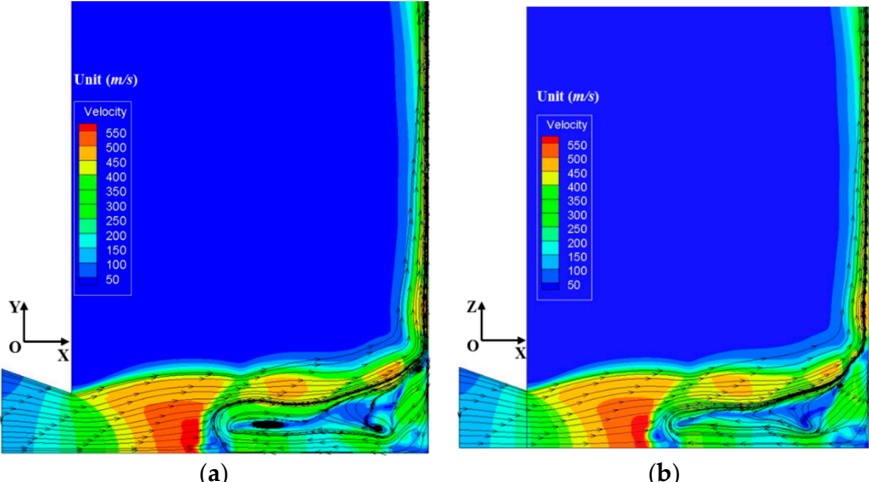

**Figure 13.** Velocity contours and streamlines at different cross-sections for gas-particle flows. (**a**) XOY plane; (**b**) XOZ plane.

Pressure map contours on ground plate for single gas flow and particle-gas flows are shown in Figures 14 and 15. Pressure was observed to be different from single gas flow and particle-gas flows in impinging jet zone. The recirculating flow led to lower pressure for particle-gas flows compared to that for single gas flow in impinging jet zone, but the maximum pressure was shown to be higher for particle-gas flows. This mainly due to that momentum of particle-gas flows was much higher than that for single gas flows, which also induced higher impinging force on the ground plate as shown in Table 1. The impinging force was calculated by Equation (10) and based on the calculated area, as shown in Figure 2. Particle mass loading of 20% was also considered. Impinging force was observed to increase with the increase of particle mass loadings. It was 1.756 and 2.044 times higher on calculated area for particle-gas flows at particle mass loading of 10% and 20% respectively compared to that for single gas flow.

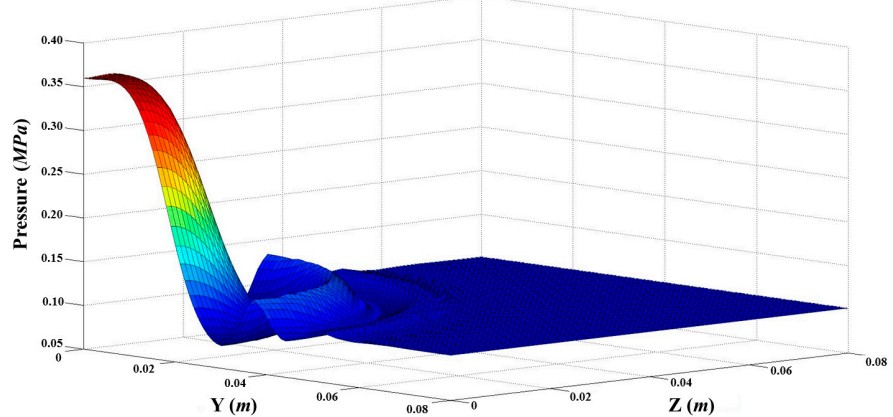

**Figure 14.** Pressure map contours on ground plate for single gas flow.

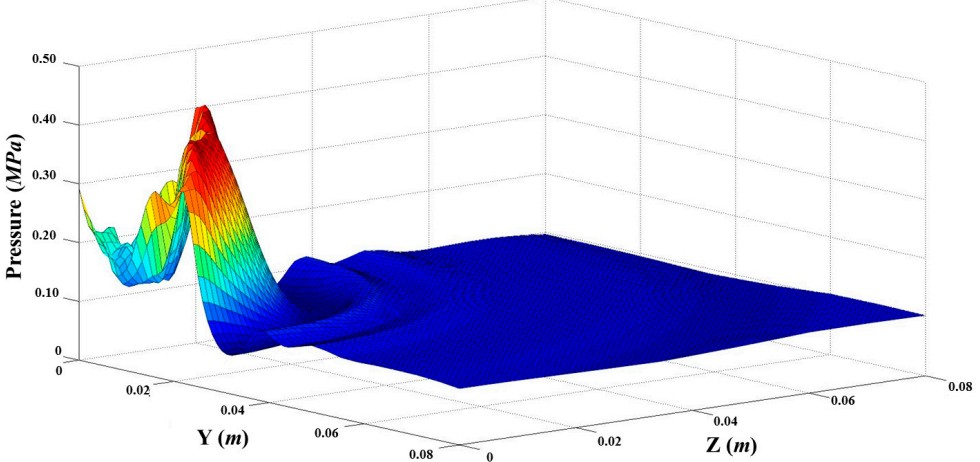

**Figure 15.** Pressure map contours on ground plate for gas-particle flows.

**Table 1.** Impinging force for single gas flow and gas-particle flows.

| Cases | Single Gas Flow | Particle Mass Loading of 10% | Particle Mass Loading of 20% |
|---|---|---|---|
| $F_I$ (N) | 109.42 | 192.14 | 223.67 |
| Non-dimensional $F_I$ | 1 | 1.756 | 2.044 |

Wall shear stress on ground plate for single gas flow and particle-gas flows is shown in Figure 16. Three flow zones were divided as stagnation flow zone, shear stress zone, and supersonic flow zone respectively. High shear stress means higher velocity gradient. In stagnation flow zone, supersonic flows became stagnation flows in impinging jet zone and wall shear stress was quite low, which can

be seen from Figures 12 and 13, where the flow velocity was low inside impinging jet zone. Due to recirculating flows occurring in impinging jet zone, there were also some high wall shear stress regions happening in stagnation flow zone for particle-gas flows. In shear layer zone, wall shear stress was the highest compared to that in other two zones. The highest shear stress layer occurred between impinging jet zone and wall jet zone, which mainly resulted from gas and particle collision. Wall shear stress was much higher in shear layer zone for particle-gas flows compared to that for single gas flow, which is mainly due to higher momentum of particle-gas flows. In supersonic flow zone, shock wave systems happened in wall jet zone and flow velocity gradient was relative higher compared to that in impinging jet zone, which also resulted to higher wall shear stress.

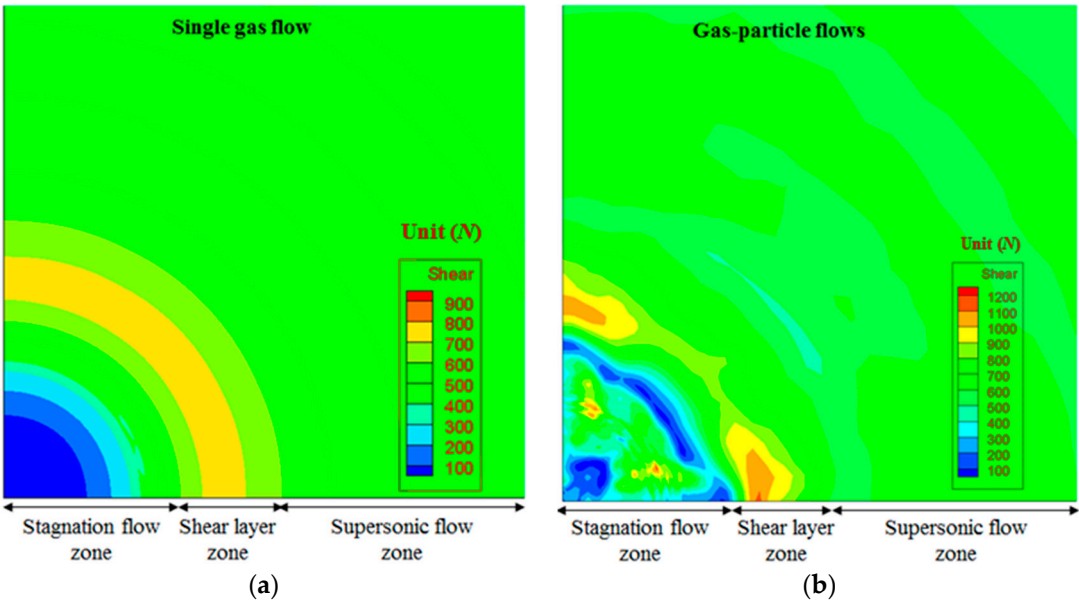

**Figure 16.** Wall shear stress contours on the ground plate for single gas flow and particle-gas flows. (**a**) Single gas flow; (**b**) Particle-gas flows.

Particle distributions induced by sonic nozzle were obtained in three-dimensional computational domain as shown in Figure 17. The quarter domain of sonic nozzle was simulated. Compared to flow velocity contours shown in Figure 13a,b, particle velocity contours indicated particles followed flow streamlines properly inside full computational domain. Particles inside primary jet zone and wall jet zone were induced to be supersonic, which also described characteristics of shock wave systems.

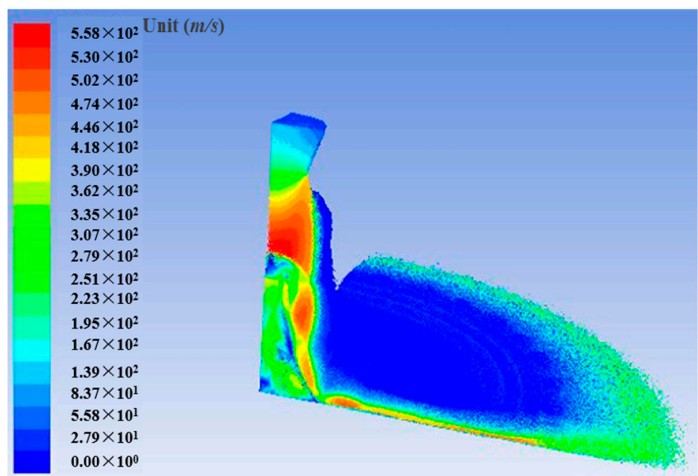

**Figure 17.** Particle velocity contours for three-dimensional domain.

## 5. Conclusions

Numerical simulations were carried out to investigate characteristics of supersonic impinging jet of particle-gas flows. Supersonic particle-gas flows induced by supersonic and sonic nozzles were studied and the effects of diameter and Stokes number of particles were also considered. Results showed that gas and particle velocity was higher from supersonic impinging jet flows induced by sonic nozzle compared to that induced by supersonic nozzle at the same operating conditions due to effects of particles. Particles at smaller diameter that results to smaller Stokes number of particles were proved to follow supersonic flow more faithfully. As Stokes number of particles increased, particle velocity gradually decreased and particles became disordered due to strong interaction between phase and particle phase. Maximum pressure and impinging force on ground plate were calculated to be higher for particle-gas flows compared to that for single gas flow, which mainly resulted from higher momentum of particle-gas flows. As particle mass loading increased, impinging force on the ground plate increased as well. Wall shear stress showed different characteristics in three flow zones. Highest shear stress zone was observed between impinging jet zone and wall jet zone, which is mainly due to gas and particle collision to the ground plate. Wall shear stress was much higher in shear layer zone for particle-gas flows compared to that for single gas flow due to higher momentum of particle-gas flows. In the present simulations, particle mass loading was fixed to be 10% and effects of particle mass loadings will be investigated in the future studies.

**Author Contributions:** Conceptualization G.Z.; methodology, G.Z.; software, Z.L.; validation, G.F.M.; formal analysis, G.Z.; investigation, G.F.M.; resources, Z.L.; data curation, G.F.M.; writing—original draft preparation, G.Z.; writing—review and editing, H.D.K.; visualization, G.Z.; supervision, Z.L.; project administration, G.Z.; funding acquisition, G.Z. All authors have read and agreed to the published version of the manuscript.

**Funding:** This research was funded by Zhejiang Sci-Tech University and National Natural Science of China, grant number 51906222.

**Acknowledgments:** The authors are grateful for the financial support of National Natural Science of China.

**Conflicts of Interest:** The authors declare no conflict of interest.

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
