# Peer review of "Numerical Analysis of Supersonic Impinging Jet Flows of Particle-Gas Two Phases"

_processes, doi:10.3390/pr8020191_

Round 1

Reviewer 1 Report

The authors have conducted unsteady RANS of a supersonic impinging jet laden with particles using DPM. The differences in the thrust and flow structures between a particle laden jet and an unladen jet is compared. The work conducted is interested and is worth of publication. However, some comments, both major and minor has to be addressed prior to the publication of the paper.

Major comments

-There are slight differences in Cp at low r/d values (figure 6). Has a grid independence study been conducted? If it has been conducted, the author should mention that the results does not change with grid refinement.

-The flow varies in the axis-symmetric plane when particles are added in the flow. Does the flow changes when half a domain or a fully domain is used instead of a quarter of a domain?  

-A supersonic flow should have a higher maximum velocity than a subsonic flow. In line 224 the author mentioned that the opposite is true. Is there an explanation to this flow phenomena?

-The figures need to be labelled (a,b, etc….) if there are 2 or more subfigures. Eg. Figures 7,9 has 4 subfigures which needs to be labelled. It is also not clear if Figures 10-12 is for a single phase flow or comparing a supersonic or sonic flow. Appropriate labels need to be used and captioned.

Minor comments

Line 41-‘As nozzle pressure ratio between nozzle inlet ….’. It is not clear when the jet is under- or over-expanded.

Figure 1. This figure needs to be references if it was taken from another source.

Line 82- Author should also mention the paper by Chan et al. IJHFF 2014 ‘Large eddy simulation and Reynolds-averaged Navier-Stokes calculations of supersonic impinging jets at varying nozzle-to-wall distances and impinging angles’

Line 88- ‘hypersonic flows more properly’ > ‘hypersonic flows more faithfully’

Line 120- equation 2<space>and

Line 127- Do you have a reference for equation (4)?

Author Response

Thank you very much for your time and effort on reviewing this article. I have given answers to all comments as shown in attachment file.

Reviewer 2 Report

Please provide a numerical grid model. In my opinion, the method used was too poorly taken into account. I expect a significant improvement in this part of the paper.

Figures 7, 9-12 - please enter some velocity units.

Figures 13 and 14 – Axis descriptions in larger font. Maybe MPa instead of Pa?

Table 1. I am sorry but I do not understand this table.

Figure15. Wall shear stress (…) - Can the shear on both slides have the same scale, for better transparency? Could the colorbar on the right slide be more readable?

Figure 16, please make it more readable. In what units are these values given?

Chapter 5. This chapter should be more extensive. You should list more results.

Author Response

Thank you very much for your time and effort on reviewing this article. I have given answers to all comments as shown in following attachment file.

Reviewer 3 Report

Paper 683795

A brief summary

Paper presents the numerical simulations of supersonic impinging jet flows of particle-gas two phases. A convergent sonic nozzle was utilized to cause supersonic impinging jet flows. The discrete phase model (DPM) with continuous phase interaction and two-way turbulence coupling model were studied and were used to simulate particle-gas flows. Gas and particle velocity contours, wall shear stress and impinging force on the ground surface were obtained to describe different phenomena inside impinging and wall jet flows of single gas phase and gas-particle two phases.

Broad comments

The work is within the scope of the journal, and the title is consistent with the content of the article. The research methodology is correct. The quality of the data and the interpretation of the analyses carried out are also good. Clear, consistent and logical conclusions were presented.

The main advantages of this paper:

Very interesting topic - 3D numerical simulations to study supersonic streams hitting particles-gas in two phases. A well-presented and illustrated numerical method supported by extensive mathematical equations.

However, in my opinion, the article requires an English native speaker consultation. In this form its a partially bit hard to reading.

Also, the number of sources cited seems too small for an article in a journal.

Specific comments

The smaller errors listed below should be corrected:

All drawing captions have no spaces between the word Figure and its current number:

For example:

Figure1 –  is; Figure 1 – should be.

Page 3

Line 120. No space.

2and 3 – is; 2 and 3 – should be.

Page 4

Line 153. No spaces.

5μm, 10μm – is; 5 μm, 10 μm – should be

Page 5

Figure 3

The “0.6d” dimension text should be slightly shifted because it overlaps with the main dimension line and makes reading difficult.

Page 6

Line 190. The legal unit of pressure in the SI system is Pa (MPa), not atm.

Author Response

Please see the attachment. Thank you very much for your time and effort on reviewing this article.

Round 2

Reviewer 2 Report

Thank you for responding to my comments.I recommend this manuscript for publication.